# Emergy and Economic Evaluation of Seven Typical Agroforestry Planting Patterns in the Karst Region of Southwest China

**Zhigang Zou [1,2,3], Fuping Zeng [1,3], Kelin Wang [1,3], Zhaoxia Zeng [1,3], Leilei Zhao [4], Hu Du [1,3], Fang Zhang [1,2,3] and Hao Zhang [1,3,5,*]**

[1]  Key Laboratory of Agro-Ecological Processes in Subtropical Region, Institute of Subtropical Agriculture, Chinese Academy of Sciences, Changsha 410125, China; zhigangzou0203@163.com (Z.Z.); fpzeng@isa.ac.cn (F.Z.); kelin@isa.ac.cn (K.W.); elizeberth@163.com (Z.Z.); hudu@isa.ac.cn (H.D.); zhangf@isa.ac.cn (F.Z.)

[2]  University of Chinese Academy of Sciences, Beijing 100049, China

[3]  Huanjiang Observation and Research Station for Karst Ecosystem, Chinese Academy of Sciences, Huanjiang 547100, China

[4]  China Forest Exploration & Design Institute in Kunming, Kunming 650216, China; zhaollsd@163.com

[5]  College of Agricultural Sciences and Natural Resources, University of Nebraska-Lincoln, Lincoln, NE 68583, USA

*   Correspondence: zhanghao@isa.ac.cn; Tel.: +186-8467-6587

**Abstract:** As a vast degraded land ecosystem, the karst region of southwest China is currently experiencing serious conflicts between restoration of degraded vegetation communities and agricultural activities. Furthermore, it is not clear what land use pattern suits local farmers best. To evaluate the sustainability of the degraded agricultural ecosystems in the region, methods for emergy analysis were used to compare the ecological and economic benefits from seven typical agroforestry planting patterns in the Yunnan province. The eco-efficiencies of the apple pattern (AP), pear pattern (PP), pomegranate pattern (PRP) were all lower than that of the traditional corn pattern (CP), although the economic benefit was higher than that of CP. Ecological benefits of the apple-soybean pattern (ASP) and the pear-pumpkin pattern (PPP) were not significantly improved, while ecological and economic benefits of the pomegranate-grass-sheep pattern (PGSP) was improved significantly. Intercropping pumpkin in PP increased the economic efficiency by 28.3%, which was superior to that of the intercropping of soybeans (4.6%) in AP. These data implied that interplanting crops in AP and PP might result in higher economic benefit than the existing interplanting pattern. The multistory agroforestry planting pattern and raising in PGSP could optimize the relationship among tree-grass-sheep and improve ecological and economic benefits. Additionally, scenario analysis showed that local farmers might enjoy better ecological and economic benefits at a large scale by optimizing current agricultural production patterns. Our results suggest that together, both the local government and farmers can adjust the structure of agroforestry ecosystems to foster the sustainable development of the ecological industry in the karst region of China.

**Keywords:** emergy analysis; agroforestry; fruit tree; degraded ecosystem; sustainable development

## 1. Introduction

Desertification is a global natural or unnatural phenomenon involving the progressive decline or loss of land productivity due to the lack of rain, vegetation destruction, wind erosion, water erosion, soil salinization [1]. As one of the most remarkable cases of desertification, rocky desertification is

common in karst regions, which cover an area of some 22 million km$^2$ that account for approximately 15% of the total land area worldwide [2]. Eroded land ecosystems in rocky desertification regions are suffering from many ecological problems, serious soil erosion, mainly, a sharp decline in forest area, rapid reduction of biodiversity and habitat poor economic development [3]. In China, the rocky desertification area comprised some 129,000 km$^2$ in 2015, including Hunan, Hubei, Guangdong, Guangxi, Guizhou, Yunnan, Chongqing and Sichuan provinces [4]. The low disaster-tolerance and environmental capacities of these areas has largely limited land use and economic development of the local population, which poses a direct threat to the ecological security of the Yangtze and Zhujiang river basins in China [5].

To reduce the rate of land degradation and to promote ecological and economic development, the government of the People's Republic of China started the nationwide comprehensive management of rocky desertification in 2008 and the Grain for Green Program (GGP) to convert cropland to grassland and forest, in 1999 [6]. Until 2017, a total area of 66,000 km$^2$ in 451 rocky desertification counties has been subjected to vegetation restoration work, fruit tree planting, forage cultivation, cattle raising, among others [7]. Many researches on rocky desertification management have evaluated a series of agroforestry ecological restoration and ecological development patterns as well, comprehensive management of agricultural and forestry systems, construction of small water-conservation projects, conversion of farmland to forest and grassland, new cooperative organizations in rural regions [8,9]. For example, Huajiang pattern emphasizes the Chinese prickly ash, dragon fruit plantation, leguminous forage grown under the plantation trees, raising livestock and poultry, development of rural biogas from animal droppings, reuse of waste residue in the biogas digester [10,11]. In order to compare the economic benefits and ecological benefits of these ecological patterns, many studies have adopted different methods for comprehensive evaluation [12–14]. However, due to the differences in evaluation methods and indicators, it is difficult to provide quantitative and unified conclusions to support the decisions of farmers or policy makers [15,16]. Therefore, it is necessary to evaluate different patterns of rock desertification by means of a unified objective quantification scheme.

Emergy analysis theory, which can convert all kinds of different categories of energy and matter to the same standard values, was first proposed for quantitative analysis in the 1990s [17]. This theory was widely used to solve the energy unit-inconformity in different nature or social systems; to measure the real value of the natural environment resources and economic activities; to analyze the relationship among components of a complex system, to harmonize ecological protection and economic and social development [18–21]. In recent decades, there have been many applications of this theory on the input-output analysis of agricultural ecosystems [22,23] to assess the level of social sustainable development [24], dynamic analysis [25,26], scenario forecasting [27]. However, most of these researches focused on the assessment of agricultural systems and farming-stockbreeding biogas systems in non-karst areas.

The karst region of southwest China is the largest land degraded ecosystem testifying to the increasingly serious conflicts between degraded vegetation restoration efforts and activities related to agricultural production [28–30]. Yunnan Province is located in the western part of this region and shows the most acute rocky desertification process in it. As is the case in the Guizhou and Guangxi provinces, the local government and people of Yunnan province have also implemented some well-known agroforestry planting ecosystems to control rocky desertification [31,32]. However, it is still unclear what pattern is best suited for the local people. In the present study, emergy and economic methods were used to evaluate these agroforestry planting patterns. Our objectives were the following: 1) to assess the emergy value of typical agroforestry planting patterns, 2) to optimize these patterns by adding ecological subsystem, and, 3) to provide the technical support for land use sustainability to the local government and farmers.

## 2. Materials and Methods

### 2.1. Research Area

The seven agroforestry planting patterns were located at Mengzi City, Honghe Hani & Yi Autonomous Prefecture, Yunnan Province, China (23°23′55″ N, 103°23′43″ E). The research area belongs to the typical karst area of faulted basins that has suffered from very serious rocky desertification. The highest and lowest elevations are 2567.8 m and 146 m, respectively. The climate belongs to the subtropical plateau monsoon climate with average annual temperature of 18.6 °C. The average annual rainfall is 815.8 mm. The annual frost-free period is 337 day, the average number of annual sunshine hours is 2234 h. Based on the statistical yearbook, government report data, our field survey data, we studied the variation of emergy values in the following agroforestry planting patterns: corn planting (CP), apple planting (AP), apple-soybean inter-planting (ASP), pear planting (PP), pear-pumpkin inter-planting (PPP), pomegranate cultivation (PRP), pomegranate-grass-sheep (PGSP) (Table 1).

**Table 1.** Background information for the seven agroforestry planting patterns.

| Pattern | Product | Location | Pre-investment | Harvest Period | Planting Area |
|---|---|---|---|---|---|
| Corn planting (CP) | Corn | Slope | Almost none | July and August | $1.33 \times 10^4$ ha |
| Apple planting (AP) | Apple | Slope | The first four years after planting require daily management and fertilizer input | July, August, September | $4.67 \times 10^3$ ha |
| Apple-soybean inter-planting (ASP) | Apple, soybean seedings | Slope | The first four years after planting require daily management and fertilizer input | July, August, September | $3.27 \times 10^2$ ha |
| Pear planting (PP) | Pear | Slope and plain | The first four years after planting require daily management and fertilizer input | July, August, September | $6.67 \times 10^3$ ha |
| Pear-pumpkin inter-planting (PPP) | Pear, pumpkin seedings | Slope | The first four years after planting require daily management and fertilizer input | July, August, September | $3.34 \times 10^2$ ha |
| Pomegranate cultivation (PRP) | Pomegranate | Bazi (middle plain of the basin) | The first five years after planting require daily management and fertilizer input | July and August | $8.33 \times 10^3$ ha |
| Pomegranate-grass-sheep pattern (PGSP) | Pomegranate, sheep | Bazi (middle plain of the basin) | The first five years after planting require daily management and fertilizer input | July and August | $8.33 \times 10^2$ ha |

### 2.2. Description of the Seven Agroforestry Planting Patterns

#### 2.2.1. Corn Planting (CP) Pattern

The Yunnan Statistical Yearbook in 2016 showed that the corn planting area is approximately 1,517,000 ha, which accounts for 21% of the total grain crop acreage in the province. Farmers plant corn on the karst soil directly and harvest it from July to August each year. This pattern is a basic planting pattern among agroforestry planting patterns.

#### 2.2.2. Apple Planting (AP) Pattern

Apple trees are planted mainly in Xibeile Town, Xin'ansuo Town, Laozhai Miao Township of Mengzi City. Total planting area has reached 6000 ha. Due to the government policy for its promotion, the apple planting area in Mengzi City is gradually expanding. Because of its prevailing special conditions, such as high altitude, low latitude, large temperature difference between day and night, sugar content in apples in Mengzi city may be as high as 17%–20%, making them some of the sweetest apples in Southwest, China. Four years after planting, apple trees begin to enter the fruiting period in the 5th year. Some early maturing apple varieties begin ripening in early July, while late maturing varieties begin to ripen in early September.

### 2.2.3. Apple-Soybean Inter-Planting (ASP) Pattern

Farmers have planted apples for more than 30 years in Mengzi City. Some farmers spontaneously plant soybeans under apple trees for extra income. Interplanting soybeans can increase economic output, while soybean roots and leaves increase soil organic matter. Intercropping also helps to reduce orchard soil erosion. Soybeans are usually sown from mid-April to May and the harvesting period comes at the end of August. Different demands in labor-intensive periods provide for full use of farmers labor.

### 2.2.4. Pear Planting (PP) Pattern

The history of pear-tree planting in Mengzi City is as long as that of apple-tree planting. The area occupied by pear-tree orchards in 2017 was about 3000 ha in Mengzi city, which is half of apple-tree orchards. According to the overall planning by the local government, the Plateau pear industry is mainly distributed in Luxi County, whose area will be more than 10,000 ha by the year 2020. The growth period of Plateau pears is about 45 days, which is shorter than that of Northern varieties, thus, has a potential advantage in the fruit market. Five years after planting, pear trees will enter the fruiting period and Plateau pears are harvested from July to August every year.

### 2.2.5. Pear-Pumpkin Inter-Planting (PPP) Pattern

Some farmers cultivate pumpkins under pear trees to increase land use efficiency and economic benefit. Pumpkin seeds are usually sown at the beginning of April and harvest takes place from August to September.

### 2.2.6. Pomegranate Cultivation (PRP) Pattern

The history of pomegranate plantation in Mengzi City dates back over 800 years. In 2017, the total area of pomegranate plantations was about 8000 ha, accounting for 48% of the orchard area in Mengzi city. It has obtained many awards by local government for its good land use efficiency and industrial development for local farmers. In recent years, sweet-seeds pomegranate with good fruit appearance and early maturity has dominated the local market. The pomegranate tree begins fruiting 10 years after planting, fruit harvest takes place in July-August.

### 2.2.7. Pomegranate-Grass-Sheep Pattern (PGSP)

To increase economic output and reduce overall economic investment, local farmers have developed an ecological engineering pattern of pomegranate-grass-livestock based on a single planted pomegranate. In this pattern, the proper amount of pasture was planted under each pomegranate tree, a certain number of stabled sheep are fed grass. Pasture can make full use of the space under the pomegranate trees, the manure produced by the sheep can be directly used as a fertilizer for pomegranates and pasture.

### 2.3. Data Collection and Sample Analysis

In this study, the questionnaire survey data, field sample data and government statistics data were mainly used. To collect basic economic data and agricultural performance of various patterns, five semi-structured questionnaire surveys on farmers, cooperatives, fertilizers, seeds, pesticide sellers for each pattern were conducted in 2017. At the same time, the soil (0–100cm) and plant samples of the corresponding farmers were collected during the questionnaire survey to determine soil bulk density, water content, total nitrogen, total phosphorus, total potassium, organic carbon content. Soil bulk density was measured by the ring knife method and soil water content was determined by the drying method. Total nitrogen and total phosphorus were digested with $H_2SO_4$ and measured by indophenol blue colorimetry and Mo-Sb colorimetry, respectively. Total potassium was measured by ICP emission spectrometry determination and total organic carbon content was measured by the

$K_2Cr_2O_7$ method [33]. In addition, the raw data on renewable natural inputs, which include incoming solar radiation, precipitation, wind speed were obtained from the weather station located at the agricultural research site. Regional agricultural acreage and development planning data were collected from the government statistics (2010–2017).

All the pre-constructions of the patterns have been completed before the research. The pre-investment per year is calculated as the ratio of pre-investment and payback period, has been considered in the economic and emergy evaluation.

### 2.4. Data Analysis

### 2.4.1. Emergy Analysis

There are three steps for emergy analysis in agroforestry planting pattern, namely drawing an aggregated system diagram, establishing emergy tables, construction of emergy-based indices. After investigating the whole system, an aggregated system diagram was drawn in accordance with the energy circuit symbols proposed by Odum in 1983 [34], which illustrates the boundaries of the system, the major components and their interactions, the materials, money and energy flows (Figure 1). On the basis of the system diagram, the emergy table is used to classify the different input and output flows, to convert the traditional units (J; g; yuan; etc.) into the unified unit (sej). Inputs are categorized as natural resources (I), purchased resources (F), system outputs (U). U is equal to the sum of yield ($Y_1$) and ecological benefits ($Y_2$). F were separated into the renewable purchased resources ($F_R$) and the non-renewable purchased resources ($F_N$). Natural resources (I) was consisted of renewable natural resources (R) and non-renewable natural resources (N). Renewable natural resources include sunlight and wind. An example of a nonrenewable natural resource is top soil loss and purchased resources include labor, fertilizer, irrigation water and seed. Emergy values are based on the benchmark calculation of $12.0 \text{ E} + 24 \text{ sej a}^{-1}$. The emergy values of all products and services were calculated using the following formula:

$$\text{Emergy (sej)} = \text{product or service (J; g; kg; \$)} \times \text{unit emergy} \left( \text{sej g}^{-1}; \text{ sej J}^{-1}; \text{ sej kg}^{-1}; \text{ sej \$}^{-1} \right)$$

All data were converted to values per ha per year. 90% of the required energy input of the required labor force is treated as non-renewable purchased resources ($F_N$) in this study, the remaining 10% is classified as renewable purchased resources ($F_R$) [20].

The final step is calculation of emergy-based indices used to assess various aspects of performance, such as resource use efficiency, environmental impact, system sustainability. Just as other studies on the sustainability of agricultural systems [35–37], emergy-power density (EPD), emergy self-sufficiency ratio (ESR), emergy yield ratio (EYR), environmental loading ratio (ELR), emergy restoration ratio (ERR), emergy benefit ratio (EBR) and emergy sustainability index (ESI) were used for balanced emergy analysis (Table 2, Table A1).

**Table 2.** Emergy evaluation indices in this research.

| Index | Function | Definition |
|---|---|---|
| Emergy-power density (EPD) | U/area | Intensity and level of economic development |
| Emergy self-sufficiency ratio (ESR) | I/U | Degree of self-sufficiency and dependence on the outside world. The autarkic ability of the system is direct proportion to the ESR. |
| Emergy yield ratio (EYR) | $Y_1/F$ | Net contribution to the economy beyond its own operation |
| Environmental loading ratio (ELR) | $(N+F_N)/(R+F_R)$ | Reflects system environmental stress and sustainability |
| Emergy restoration ratio (ERR) | $Y_2/F$ | Ecological benefits of rocky-desertification control |
| Emergy benefit ratio (EBR) | $(Y_1+Y_2)/F$ | Ecological and economic benefits of rocky-desertification control |
| Emergy sustainability index (ESI) | EYR/ELR | Sustainability of the system. ESI value ranged from 1 to 10, is direct proportion to the sustainability of the production system. |

Note: I: Natural Resources. R: Renewable natural resources. N: Non-renewable natural resources. F: Purchased resources. $F_R$: Renewable purchased resources. $F_N$: Non-renewable purchased resources. U = I + F. $Y_1$ = U, system outputs. $Y_2$, ecological benefits for water conservation (WC), soil reinforcements (SR) and fertility (FE), carbon fixation (CF), oxygen production (OP), $Y_2$ = WC + SR + FE + CF + OP.

### 2.4.2. Economic Analysis

The economic output/input ratio, unit economic benefit (UEB), unit net economic benefit (UNEB) were used for economic analysis (TableA2). Economic output/input ratios can be used for measurement of economic cost efficiency, while UEB and UNEB are used for the measurement of the economic benefits from the whole agriculture ecosystem.

$$UEB = O - I,$$

$$UNEB = O - I_a,$$

$$I = I_a + I_f,$$

O: System economic output. I: System economic input. $I_a$: the actual economic input of the system, $I_f$: the free input of the system, which is equal to the market value of the labor and manure input of farmers themselves.

### 2.4.3. Scenario Analysis

Scenario analysis is a quantitative and qualitative combination method [38] used to compare and study the possible status under different trend conditions. Scenario analysis can predict the long-term outcome of studies with clear-cut factors and is well suited to predicting the status of policy or to measure implementation years later [39]. In this research, we constructed a baseline scenario (BS), an improvement scenario (IS), an optimization scenario (OS) based on field research data, expert experience and relevant plans of the Mengzi Municipal Government (Table 3). The ratio of area conversion in the optimization scenario was calculated by the maximum value in the implemented plan. All ratios of these conversion were optimized by the recommendations of experts.

**Table 3.** Setting and comparing of different scenarios.

| Scenarios | Definition | Status after 5 years |
|---|---|---|
| Baseline scenario | Government and business have not adjusted the existing promotion and subsidy policies; the regional planting structure has not changed significantly. There is no major technological change in farming methods. | Pomegranate planting area is 10,000 ha, 10% of which is PGSP. Apple planting area is 6330 ha, 7% of which is ASP. Pear planting area is 3000 ha, 5% of which is PPP. Corn planting area is 13,670 ha. |
| Improvement scenario | Government and business have promoted the transformation of the planting structure, especially the transition from original planting patterns to ecological planting patterns. | Pomegranate planting area is 10,600 ha, 15% of which is PGSP. Apple planting area is 7000 ha, 10% of which is ASP. Pear planting area is 3000 ha, 7% of which is PPP. Corn planting area is 12,400 ha. |
| Optimization scenario | Based on the improvement scenario, the government has promoted the transformation of corn cultivation to pomegranate and apple, actively promoted the transformation of the traditional patterns to the ecological patterns. | Pomegranate planting area is 11,330 ha, 30% of which is PGSP. Apple planting area is 8670 ha, 15% of which is ASP. Pear planting area is 3000 ha, 10% of which is PPP. Corn planting area is 10,000 ha. |

## 3. Results

### 3.1. Emergy Analysis of Seven Typical Agroforestry Planting Patterns

The traditional CP has the simplest structure of emergy systems in these patterns (Figure 1). With the addition of subsystems, the emergy system structure of PGSP became more complex than before. Except for CP and PGSP, the other five emergy systems showed similar structure. The proportions of seed, sheep, fertilizer and pesticide in all planting patterns were very large. The manure and compound fertilizer in CP accounted for 45.5% and 32.7% of total investment, respectively, which accounted for 78.2% of the grand total (Figure 2). The fertilizer input in the PRP also accounts for an absolute dominant position. In addition, over 60% of fertilizer consumption in PP, PPP, AP, ASP is mainly as compound fertilizers; 94.6% of fertilizer consumption in PRP pattern is in the form of organic fertilizer. When grass is added to the sheep subsystem, the proportion of fertilizer dropped to 30.7% and the input ratio of seeds and mutton increased from 0.66% to 64.5%. Thus, total emergy input in PRP was $2.67{\times}E + 17$ sej$^{-1}{\cdot}$ha$^{-1}{\cdot}$year$^{-1}$, while that in CP was $1.69{\times}E + 16$ sej$^{-1}{\cdot}$ha$^{-1}{\cdot}$year$^{-1}$. Subsystem addition to PRP reduced total input of emergy by 19.6%, which indicated that the addition of subsystem significantly reduces the input requirements for external emergy in PRP. The emergy input in AP ($2.16{\times}E + 16$ sej${\cdot}$hm$^{-1}{\cdot}$a$^{-1}$) was slightly higher than that of PP ($1.83{\times}E + 16$ sej${\cdot}$hm$^{-1}{\cdot}$a$^{-1}$). Similarly, it was different in the impact of the interplanting and grass-sheep subsystem on PP. PPP and ASP increased the demand for external emergy by 6.6% and 4.0%, respectively.

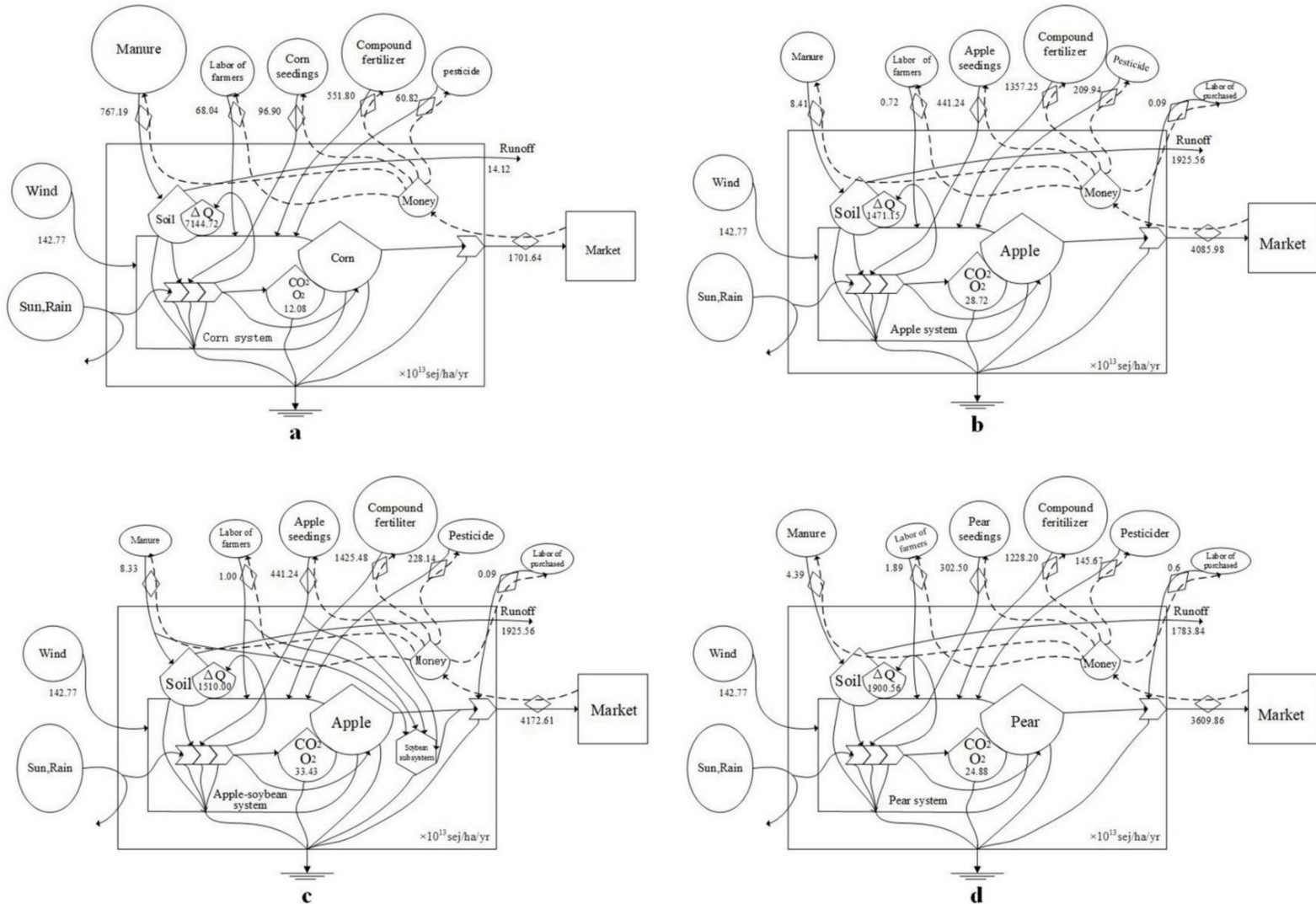

**Figure 1.** *Cont.*

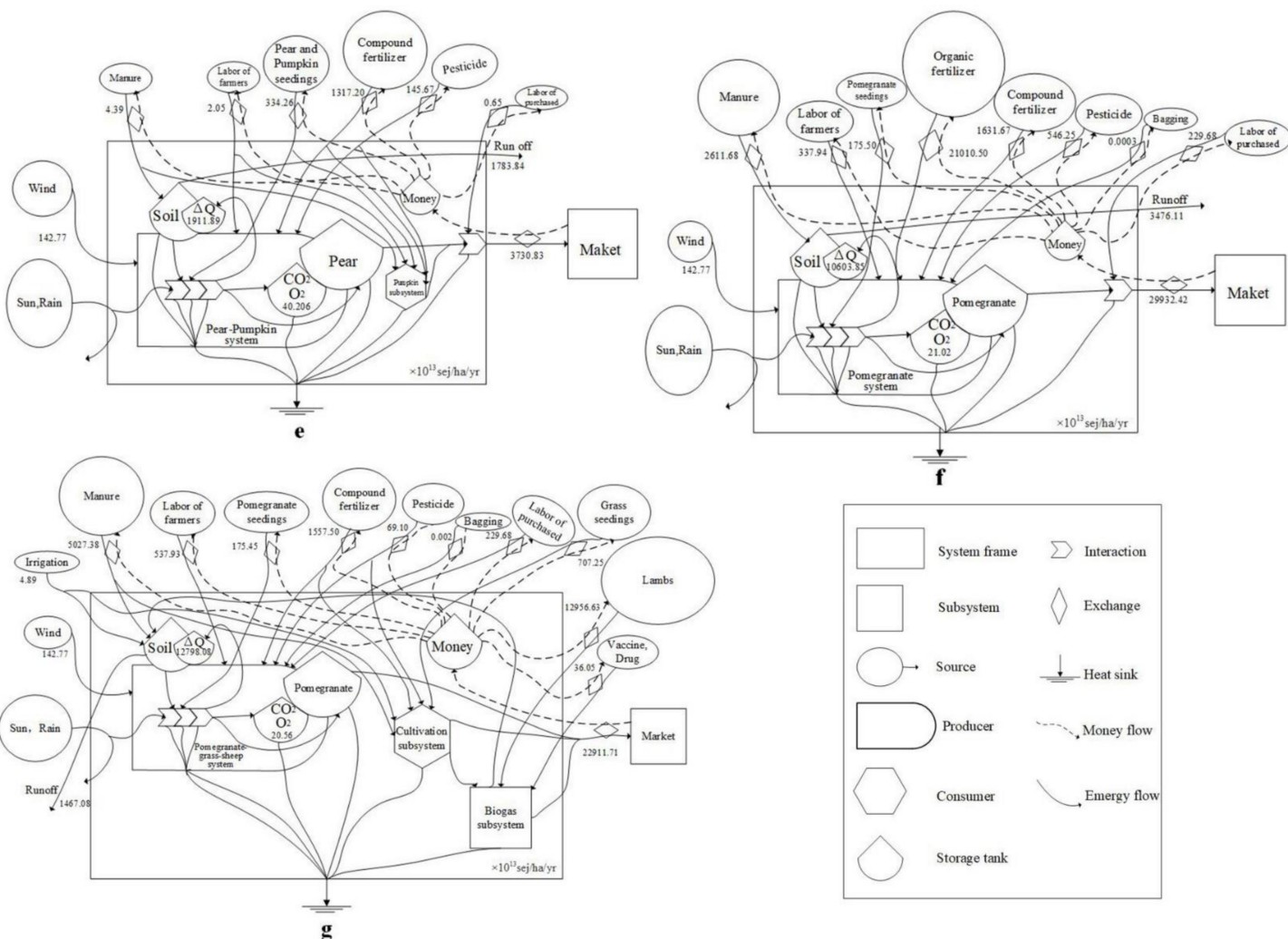

**Figure 1.** The emergy systems of seven typical agroforestry planting. **a**: Corn planting (CP) pattern. **b**: Apple planting (AP) pattern. **c**: Apple-soybean inter-planting (ASP) pattern. **d**: Pear planting (PP) pattern. **e**: Pear-pumpkin inter-planting (PPP) pattern. **f**: Pomegranate cultivation (PRP) pattern. **g**: Pomegranate-grass-sheep pattern (PGSP).

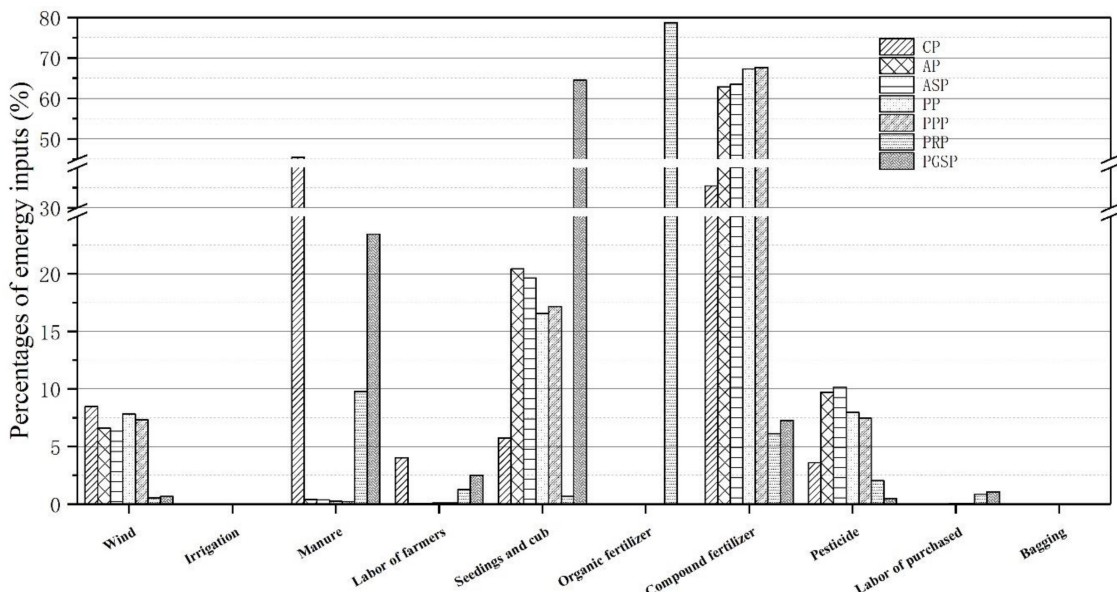

**Figure 2.** Emergy inputs percentages of the seven typical agroforestry planting patterns.

CP showed the lowest emergy-power density (EPD) ($18.20 \times$ E + 11 sej$^{-1} \cdot$ha$^{-1} \cdot$year$^{-1}$) among all patterns under study, while PRP showed the highest EPD ($296.54 \times$ E + 11 sej$^{-1} \cdot$ha$^{-1} \cdot$year$^{-1}$). The impact of the intercropping system on the original planting pattern was not obvious. However, the addition of the grass-sheep subsystem had a significant effect on the emergy density in PRP.

As shown in Table 4, the emergy self-sufficiency ratio ESR ranged from 0.08 to 0.58. CP, PRP, PGSP required more emergy than the other patterns. ESR of CP was also very low, which showed that both CP and PGSP required an external input. Because of the small effect of the addition of intercropping plants on the energy structure, ESR values for AP and ASP, PP and PPP were quite similar. ESR for PGSP was 33.3% lower than that for PRP.

**Table 4.** Emergy evaluation in the seven agroforestry planting patterns.

| Indices | CP | AP | ASP | PP | PPP | PRP | PGSP |
|---|---|---|---|---|---|---|---|
| Emergy-power Density (EPD) ($\times$E+11) | $18.20 \pm 4.92$ | $42.04 \pm 18.26$ | $42.91 \pm 17.51$ | $37.38 \pm 4.66$ | $38.59 \pm 3.98$ | $296.54 \pm 11.53$ | $220.74 \pm 50.88$ |
| Emergy Self-sufficiency Ratio (ESR) | $0.15 \pm 0.05$ | $0.58 \pm 0.20$ | $0.56 \pm 0.19$ | $0.55 \pm 0.06$ | $0.53 \pm 0.05$ | $0.12 \pm 0.01$ | $0.08 \pm 0.02$ |
| Emergy Yield Ratio (EYR) | $0.99 \pm 0.06$ | $2.70 \pm 1.03$ | $2.51 \pm 0.87$ | $2.22 \pm 0.34$ | $2.14 \pm 0.25$ | $2.41 \pm 0.58$ | $3.52 \pm 0.82$ |
| Environmental loading Ratio (ELR) | $1.30 \pm 0.0.05$ | $14.63 \pm 6.34$ | $14.96 \pm 6.06$ | $12.69 \pm 1.09$ | $13.13 \pm 0.92$ | $13.80 \pm 2.26$ | $4.95 \pm 0.93$ |
| Emergy Restoration Ratio (ERR) | $4.64 \pm 1.78$ | $1.10 \pm 0.60$ | $1.04 \pm 0.55$ | $1.14 \pm 1.03$ | $1.08 \pm 0.89$ | $0.41 \pm 0.01$ | $0.62 \pm 0.15$ |
| Emergy Benefit Ratio (EBR) | $5.63 \pm 1.80$ | $3.80 \pm 1.64$ | $3.55 \pm 1.42$ | $3.36 \pm 1.36$ | $3.22 \pm 1.14$ | $2.82 \pm 0.58$ | $4.17 \pm 0.89$ |
| Emergy Sustainability Index (ESI) | $0.76 \pm 0.02$ | $0.22 \pm 0.13$ | $0.20 \pm 0.11$ | $0.18 \pm 0.04$ | $0.16 \pm 0.03$ | $0.18 \pm 0.06$ | $0.74 \pm 0.28$ |

The emergy yield ratio (EYR) ranged from 0.99 to 3.52 across patterns (Table 4); ESR for AP was 17.8% higher than that for PP, while the grass-sheep subsystem increased EYR by 46.1%.

The environmental loading ratio (ELR) for CP and PGSP were lowest and highest across patterns, respectively (Table 4); ELR values for AP, ASP, PP, PPP, PRP demonstrated that these patterns depended heavily on non-renewable resources and low utilization of renewable resources, which may pose a great pressure on the environment. Adding the grass-sheep subsystem to PRP can significantly reduce as much as 64.1% of the environmental load.

The highest and lowest values for the emergy restoration ratio (ERR) among all patterns were found for CP and PRP, respectively (Figure 3). The addition of a subsystem to PRP increased ERR

by 51.2%; ERR calculated for PGSP (0.64) was much lower than for CP (4.64), which indicated that the control effect of rocky desertification was very unsatisfactory. The control benefits of rocky desertification in the other patterns do not seem to be as good as in CP.

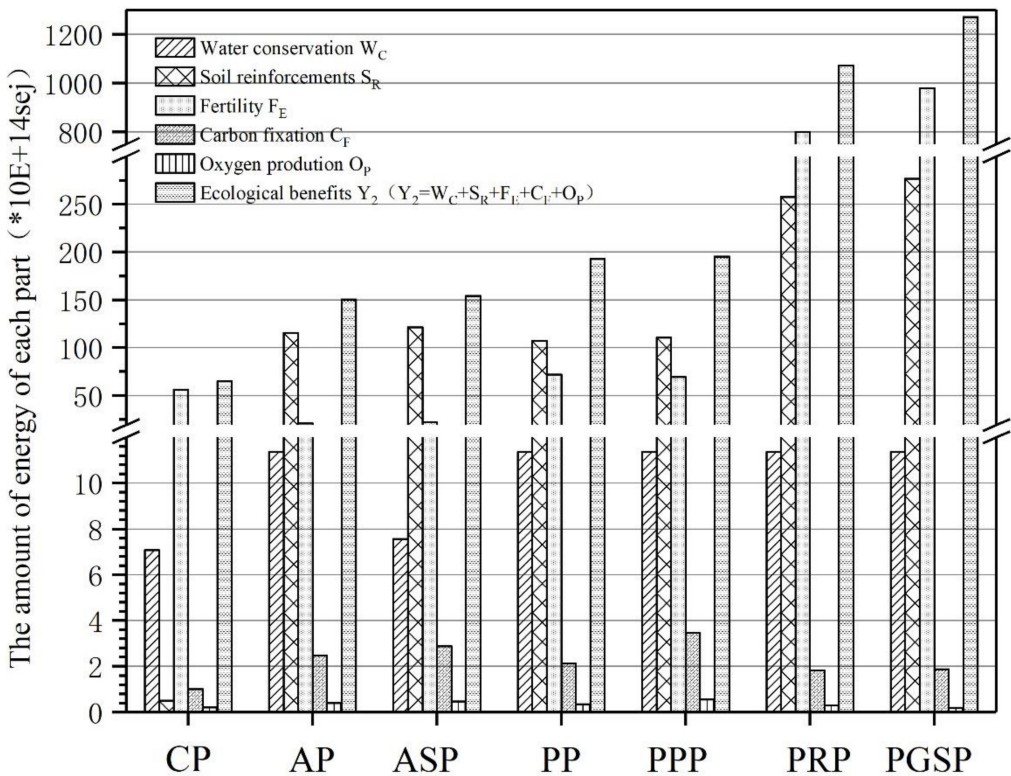

**Figure 3.** Ecological benefits of the seven typical agroforestry planting patterns.

The emergy benefit ratio (EBR) for CP (5.63) was the highest among all patterns studied, while that for PRP (2.82) was the lowest. Calculated EBR values for AP and PP were 3.80 and 3.36, respectively, which was also higher than that of PRP. The addition of intercropped plants in AP and PP slightly reduced EBR. However, the addition of the grass-sheep subsystem in PRP significantly increased EBR by up to 47.9%, which may reduce the external inputs and result in high ecological efficiency and emergy output.

The emergy sustainability index (ESI) can be broadly divided into two categories for all seven patterns under study here; these include higher ESI values for CP and PGSP, lower ESI values for AP, ASP, PP, PRP. Compared with the simple planting pattern, the intercropping pattern had little effect on ELR and EYR. The addition of the grass-sheep subsystem to PRP enhanced EYR and significantly reduced ELR. The ESI in PGSP increased by 311.1% over that of PRP.

*3.2. Economic Benefit from the Seven Typical Agroforestry Planting Patterns*

Irrespective of free input, economic input from highest to lowest ranked as follows: PRP > PGSP > ASP > PPP > AP > PP > CP (Table 5). On the other hand, if free input is taken into consideration, the ranking is as follows: PGSP > PRP > PPP > PP > ASP > AP > CP.

**Table 5.** Indices for the economic evaluation of the seven agroforestry planting patterns.

| Indices | CP | AP | ASP | PP | PPP | PRP | PGSP |
|---|---|---|---|---|---|---|---|
| Input with $I_f$ (I) (yuan year$^{-1}$ ha$^{-1}$) | 27,145.83 | 28,034.67 | 30,521.33 | 32,374.33 | 34,386.83 | 164,662.50 | 212,679.33 |
| Input without $I_f$ ($I_a$) (yuan year$^{-1}$ ha$^{-1}$) | 5379.17 | 24,140.67 | 25,324 | 23,183.33 | 24,445.83 | 83,908.33 | 64,276.83 |
| Output (O) (yuan year$^{-1}$ ha$^{-1}$) | 12,650.00 | 111,890.00 | 117,033.33 | 107,083.33 | 13,7402.08 | 520,000.00 | 583,250.00 |
| Economic Output /Input with $I_f$ (O/I) | 0.46 ± 0.06 | 4.72 ± 1.22 | 4.25 ± 0.80 | 3.33 ± 0.71 | 4.03 ± 0.80 | 3.15 ± 0.78 | 2.72 ± 0.52 |
| Economic Output/Input without $I_f$ (O/I) | 2.36 ± 0.12 | 5.5 ± 1.43 | 5.23 ± 1.08 | 4.73 ± 1.22 | 5.77 ± 1.43 | 6.17 ± 1.53 | 9.04 ± 1.98 |
| Unit economic benefit (EBU, O-I) (yuan year$^{-1}$ ha$^{-1}$) | 7270.8 ± 2406.25 | 83,855.33 ± 46,050.60 | 86,512 ± 46,761.43 | 74,709 ± 24,596.39 | 103,015.25 ± 26,618.89 | 355,337.5 ± 131,544.00 | 370,570.67 ± 130,768.10 |
| Unit net economic benefit (EPBU, O-$I_a$) (yuan year$^{-1}$ ha$^{-1}$) | 14,495.83 ± 2296.48 | 87,749.33 ± 49,540.65 | 91,709.33 ± 50,109.60 | 83,900 ± 25,640.93 | 112,956.25 ± 27,539.48 | 436,091.67 ± 136,716.63 | 518,973.17 ± 141,696.23 |

Note: O is the economic output of the systems. I is the economic input of the systems. $I_a$ is the actual economic input of the systems, $I_f$ is the input for free of the systems (the labor from the farmers and the poultry manure from the breeding subsystem) which was converted to the market value. $I = I_a + I_f$.

Economic output from CP was lowest (1265.00 yuan·year$^{-1}$ ha$^{-1}$) among all patterns studied, representing only 11.8% of PP and 2.2% of PGSP. When free investment was considered, output from CP was 46.6% of the corresponding input. With increasing investment, the economic output from PPP was 28.3% higher than that from PP, while the output from ASP increased by 4.6% over that from AP. Compared to PP, PGSP reduced external investment in purchases and increased input of farmer labor and free fertilizer by 12.2% of the output. The interplanting of pumpkin and the addition of the grass-sheep subsystem improved economic benefit (Table 5).

Compared to AP, ASP shows a reduced Economic Output/Input when free investment is considered. Conversely, compared to PP, PPP showed an increased Economic Output/Input when free investment was considered. Compared to PRP, the Economic Output/Input for PGSP was reduced by 13.6% when free input was considered. When free input was not considered, the Economic Output/Input for PGSP increased by 46.5%. In both cases, CP showed the lowest Economic Output/Input among the seven patterns. In this case, when free investment was considered, the Economic Output/Input was only 0.46, which indicated that the economic revenue from CP was mainly the marketing return for free farmer labor and free fertilizer. The observed pattern ranking based on Economic benefit per unit and Economic pure benefit per unit was the following: PGSP > PRP > PPP > ASP > AP > PP > CP.

*3.3. Scenario Analysis of Seven Typical Agroforestry Planting Patterns*

Based on the scenario setting and pattern evaluation, the economic output is the major concern in the government promotion plan (Table 6). In the basic scenario, the total economic benefit from cultivation of pomegranate, corn, apple and pear in Mengzi City would increase by 8.8%, while the total ecological benefits remain unchanged. In the improvement scenario, the corn planting area would be transformed into pomegranate and apple orchards, while the transformation of PRP, AP and PP into ecological planting patterns would be promoted. The total economic benefit would increase by 16.1% while the total ecological benefits would increase by 6.2%. In this process, the corn planting area would decrease from 13,333 ha to 12,400 ha, (−7%). Finally, in the optimization scenario, the improvement of economic benefits and ecological benefits would be significant. In the promotion plan of the planting pattern, the area of all planting patterns would increase, while these patterns would all have better economic benefits. The current government promotion plan would not significantly improve the ecological benefits. With an increase of ecological planting patterns, the economic and ecological benefits would be further improved. The patterns under study here cannot improve the ecological and economic benefits equally.

**Table 6.** Emergy and economic status after five years under each scenario constructed.

| Scenario | Total planting area (ha) | Total net economic benefit (yuan year$^{-1}$ ha$^{-1}$) | EACI (%) | Total economic benefit (yuan year$^{-1}$ ha$^{-1}$) | EACI (%) | Total ecological benefit (sej year$^{-1}$ ha$^{-1}$) | EACI (%) |
|---|---|---|---|---|---|---|---|
| Current situation | 30,799 | 3,527,170,867 | | 4,600,534,572 | | $1.37953 \times 10^{21}$ | |
| Baseline scenario | 33,000 | 4,136,793,863 | 9.4 | 5,362,392,475 | 8.8 | $1.5866 \times 10^{21}$ | 0.2 |
| Improvement scenario | 33,000 | 4,435,287,213 | 17.4 | 5,725,326,213 | 16.1 | $1.68236 \times 10^{21}$ | 6.2 |
| Optimization scenario | 33,000 | 4,900,887,703 | 29.7 | 6,326,987,305 | 28.4 | $1.85222 \times 10^{21}$ | 17.0 |

EACI: Equal area change of the index. In the current government planning adjustment, the planting area has increased. In order to eliminate the numerical difference caused by the increase in area, the indicators are specifically converted into the current area for comparison.

## 4. Discussion

The results presented herein showed that the four basic patterns (CP, AP, PP, PRP) were different in performance, according to emergy analysis (Tables 3 and 4). The AP pattern showed lower EPD, EYR, ELR, higher ESI, which is in stark contrast to the PRP pattern. While the former represents a traditional pattern with less manual interference, low input, low output, low pressure, the pomegranate market is competitive, prices vary greatly. Local farmers had to become more industry-oriented while planting pomegranate, thus resorting to new practice, such as bagging, investing more in labor and non-renewable resources. Although different types of resources and input forms were required in these patterns, labor and fertilizer dominated the major contribution to the total input, which was consistent with the China's crop production system [40].

The emergy for the CP pattern in the study area differed from other studies on corn in Yunnan [41]. EYR and ELR were higher than the corresponding values for Danish traditional feed corn production method [42], for the Guangxi Duan Agricultural Ecological Economic System in the karst area [43]. The difference may be due to the more serious rocky desertification and the lower planting density in our study area. The ELR for the PRP pattern was much higher than that in the Duan area, which in turn is higher than that in Guizhou Province [44]. This difference may due to the long history of pomegranate cultivation in the study area, the large single-headed planting area, the reasonable level of management and fertilization. The gap between the AP and the PP patterns is very small. The overall emergy and the economic indicator for AP are slightly better than those for PP. The EYR value obtained by Zhang et al. [45,46] in Dengcheng County in Shaanxi Province was relatively consistent with 2.69, while the ELR value calculated in this study was much higher than that by Zhang et al. for a different apple planting density and another apple genotype.

In compound agroforestry planting patterns, the contribution from both and PPP to the original respective patterns were relatively limited. The emergy indicators for these two patterns decreased, because the intercropping pattern requires more input of fertilizers and pesticides. The utilization efficiency of natural resources of the whole system decreased, the shading of light make the less growth in intercropping crops than that in monoculture [47]. The economic indicators of ASP and PPP showed that the intercropping of pear and pumpkin had a better collaborative utilization of resources. Comparing with PRP, the emergy of PGSP was significantly improved, which was similar as the previous report [48]. Except for the deterioration of the ESR, other indicators have greatly improved. Compared with Pig-Marsh-Pomegranate pattern in the rural area of Linyi City, Shandong Province [47], ESR, ELR, EYR calculated here were lower because the biogas system contributes to the recycling of manure and urine at a high resource utilization rate. At the same time, local government subsidy to Pig-Marsh-Pomegranate pattern reduces investment. Compared with the overall evaluation of results in Shandong Province, the EYR index for all patterns under study here was lower than, because the conditions for agricultural production in the karst region are poorer than in non-karst regions [49].

The ELR values calculated here for AP, ASP, PP, PPP, PRP were all higher than the overall level for Yunnan Province [50], which indicated that, in this study, these patterns showed higher pressure on the environment than the average load in Yunnan Province. Our data is inconsistent with the orientation of the agricultural planting pattern and needs to be improved as soon as possible. The ELR values for CP and PGSP are lower than the overall level for Yunnan Province, the pressure on the environment is appropriate. In addition, the values of ESI are similar to that of ELR. The ESI values for AP, ASP, PP, PPP, PRP are much lower than that for the overall level in Yunnan Province, while ESI values for CP and PGSP are slightly higher than overall level in the province.

Compared to the evaluation of national energy production (maize, wheat, rice, soybean; EYR > 1.45, ELR > 4.50, ESI < 0.33), the EYR and ELR values for CP in this study were lower than the corresponding values for these other four grain production patterns in 2015, while the ESI value was much higher [51]. This difference may be due to the influence of the study area and karst topography, the scale of corn cultivation, on EYR and ELR values. At the same time, the ESI value is consistent with those of maize, wheat and rice, while both, EYR and ELR are approximately twice the value for maize, wheat, rice at national level. Compared with simple grain production patterns, fruit cultivation has a relatively high-input and high-output pattern, better emergy performance. EYR, ELR and ESI values of PGSP were twice as high as those of soybean, indicating that PGSP was superior to all grain production patterns in terms of emergy performance. In addition, ESI, EYR of PGSP was significantly lower than that of biogas-linked pattern [47,52], while EYR was higher than that of greenhouse linked pattern [53]. However, due to the lack of analysis of specific economic indicators in previous studies, it is impossible to make a more comprehensive comparison of economic and ecological benefits with this study. As far as results from our study are concerned, PGSP achieved very good performance, while AP, ASP, PP, PPP, PRP currently show some deficiencies in terms of emergy and economic performance that need to be corrected.

The overall performance of corn in the karst region in Yunnan Province is poor; therefore, we propose that the government should advocate the reduction of corn planting. Additionally, although PPP and ASP perform better than corn, the local government needs to introduce appropriate breeding systems to optimize current ASP and PPP [54]. Similarly, PGSP shows attractive ecological and economic benefits, it is still necessary to improve the ratio value for a scientifically standardized aquaculture subsystem in the existing PGSP. Castellini's research on two poultry farms in Italy showed that the adding of grazing to traditional orchard was similar with PGSP [55]. At the same time, it is desirable to attempt the introduction of a biogas system to reuse the increasing stock waste [56,57]. Because apples, pears, pomegranates together comprise a relatively long harvest period, the local government should also coordinate the establishment of a temporary labor market to meet the demand for labor.

Our scenario analysis allowed a better understanding of potential ecological and economic benefits under a better planting structure configuration. According to current government plans, after five years, the economy and ecology of the different agroforestry planting patterns studied here will have increased differentially. Therefore, we suggest that the government makes greater efforts in the allocation and improvement of these patterns.

Additionally, although our research has drawn some suggestions on these seven patterns, we still need to carry out multi-year observation on the sample land in multiple regions [58,59], In the future research, the geographic information system, emergetic ternary diagrams and other methods may be used to guide the pattern improvement and agricultural planning at regional scale [60]. At the same time, there is still a need to investigate ecosystem sustainability, driving force and biodiversity change at different scales combining existing models [61–63].

Overall, emergy analysis used in this study made full consideration of all aspects, from agricultural production to consumption and utilization of natural resources; thus, it included the more objective input and output analysis of the agricultural production process and hence, a more reasonable sustainability analysis of each ecological economic system [64]. Further, we realize that the actual

energy conversion rates may be biased for the differences in natural conditions, as well as social and economic development and productivity levels [65]. Thus, future research should focus on the ecological economic system of energy-conversion efficiency in the karst region for proper correction of available energy-conversion rates to assess the ecological systems and possible economic development.

## 5. Conclusions

At the ecosystem scale, the economic benefits of the remaining six agroforestry planting patterns were significantly better than the traditional corn pattern. The ecological benefits of AP and PP did not increase after intercropping was added. After added subsystem of grass + sheep to these patterns, the ecological and economic benefits of PGSP were obviously improved. At the regional scale, scenario analysis indicated that the existing government planning is purely economic oriented. After the transformation of existing patterns to ecological agroforestry planting patterns (ASP, PPP and PGSP), the government and farmers can get better economic and ecological benefits.

**Author Contributions:** Conceptualization, H.Z. and Z.Z. (Zhigang Zou); methodology, K.W. and F.Z. (Fuping Zeng); investigation, H.Z., H.D., F.Z. (Fang Zhang) and Z.Z. (Zhaoxia Zeng); writing—original draft preparation, H.Z. and Z.Z. (Zhigang Zou); writing—review and editing, H.Z. and Z.Z. (Zhigang Zou); funding acquisition, F.Z. (Fuping Zeng).

**Funding:** This research was funded by the National Key Research and Development Program (2016YFC0502505, 2016YFC0502400); National Science and Technology Support Plan (2015BAD06B04); National Natural Science Foundation of China (31870712); Guangxi Innovation-driven Development Program (AA18118015), Guangxi Key Research and Development Program (AB17129002, AB17292064, AB17129009), Innovation Research Team Project of Institute of Subtropical Agriculture, Chinese academy of sciences (2017QNCXTD_XXL) and Guangxi Provincial Program of Distinguished Experts in China.

**Acknowledgments:** We acknowledge the technical and field work support from Jianyun Zhang, Jiejun Zhao, Songlian Bao.

**Conflicts of Interest:** The authors declare no conflict of interest.

# Appendix A

**Table A1.** Emergy analysis table of the CP in 2017(/ha/year).

| Item | Raw Amounts | | | EUVs (sej unit$^{-1}$) | Solar Emergy (sej) | | | Average (sej) |
|---|---|---|---|---|---|---|---|---|
| | 1 | 2 | 3 | | 1 | 2 | 3 | |
| **Input** | | | | | | | | |
| Renewable resource (R) | | | | | | | | |
| Solar radiation (J) | $5.09 \times 10^{13}$ | $5.09 \times 10^{13}$ | $5.09 \times 10^{13}$ | 1 | $5.09 \times 10^{13}$ | $5.09 \times 10^{13}$ | $5.09 \times 10^{13}$ | $5.09 \times 10^{13}$ |
| Wind (J) | $7.51 \times 10^{11}$ | $7.51 \times 10^{11}$ | $7.51 \times 10^{11}$ | 1900 | $1.43 \times 10^{15}$ | $1.43 \times 10^{15}$ | $1.43 \times 10^{15}$ | $1.43 \times 10^{15}$ |
| Rain (chemical) (J) | $4.83 \times 10^{10}$ | $4.83 \times 10^{10}$ | $4.83 \times 10^{10}$ | 23,500 | $1.13 \times 10^{15}$ | $1.13 \times 10^{15}$ | $1.13 \times 10^{15}$ | $1.13 \times 10^{15}$ |
| Water (for irrigation) (J) | | | | | | | | |
| Subtotal, R = Rain + Water | | | | | $2.61 \times 10^{15}$ | $2.61 \times 10^{15}$ | $2.61 \times 10^{15}$ | $2.61 \times 10^{15}$ |
| Nonrenewable resource (N) | | | | | | | | |
| Loss of topsoil | | | | | | | | |
| Total N (g) | 2580 | 2580 | 2580 | 464,000,000 [18] | $1.20 \times 10^{12}$ | $1.20 \times 10^{12}$ | $1.20 \times 10^{12}$ | $1.20 \times 10^{12}$ |
| Total P (g) | 2280 | 2280 | 2280 | $5.07 \times 10^{9}$ [18] | $1.16 \times 10^{13}$ | $1.16 \times 10^{13}$ | $1.16 \times 10^{13}$ | $1.16 \times 10^{13}$ |
| Total K (g) | 58,000 | 58,000 | 58,000 | $1.31 \times 10^{9}$ | $7.60 \times 10^{13}$ | $7.60 \times 10^{13}$ | $7.60 \times 10^{13}$ | $7.60 \times 10^{13}$ |
| Organic (J) | 557,000,000 | 557,000,000 | 557,000,000 | 94,100 [66] | $5.24 \times 10^{13}$ | $5.24 \times 10^{13}$ | $5.24 \times 10^{13}$ | $5.24 \times 10^{13}$ |
| Subtotal (Total N + Total P + Total K + Organic) | | | | | $1.41 \times 10^{14}$ | $1.41 \times 10^{14}$ | $1.41 \times 10^{14}$ | $1.41 \times 10^{14}$ |
| Local resource (I), I = R + N | | | | | $2.75 \times 10^{15}$ | $2.75 \times 10^{15}$ | $2.75 \times 10^{15}$ | $2.75 \times 10^{15}$ |
| Purchased renewable resource (F$_R$) | | | | | | | | |
| Labor (10%)$^{\#}$ (J) | 21,800,000 | 58,000,000 | 13,100,000 | 2,200,000 [67] | $4.79 \times 10^{13}$ | $1.28 \times 10^{14}$ | $2.87 \times 10^{13}$ | $6.81 \times 10^{13}$ |
| Manure (68%)$^{\#\#}$ (J) | $1.79 \times 10^{11}$ | $7.67 \times 10^{10}$ | $1.92 \times 10^{11}$ | 35,000 | $6.26 \times 10^{15}$ | $2.68 \times 10^{15}$ | $6.71 \times 10^{15}$ | $5.22 \times 10^{15}$ |
| Subtotal | | | | | $6.31 \times 10^{15}$ | $2.81 \times 10^{15}$ | $6.74 \times 10^{15}$ | $5.28 \times 10^{15}$ |
| Purchased nonrenewable resource (F$_N$) | | | | | | | | |
| Seeds of corn (yuan) | 825 | 750 | 825 | $1.21 \times 10^{12}$ | $9.98 \times 10^{14}$ | $9.08 \times 10^{14}$ | $9.98 \times 10^{14}$ | $9.68 \times 10^{14}$ |
| Labor (90%)$^{\#}$ (J) | $1.96 \times 10^{8}$ | $5.22 \times 10^{8}$ | $1.17 \times 10^{8}$ | 2,200,000 [67] | $4.31 \times 10^{14}$ | $1.15 \times 10^{15}$ | $2.58 \times 10^{14}$ | $6.12 \times 10^{14}$ |
| Compound fertilizer (kg) | 1650 | 1000 | 2000 | $3.56 \times 10^{12}$ [17] | $5.87 \times 10^{15}$ | $3.56 \times 10^{15}$ | $7.12 \times 10^{15}$ | $5.52 \times 10^{15}$ |
| Manure (32%)$^{\#\#}$ (J) | $8.42 \times 10^{10}$ | $3.61 \times 10^{10}$ | $9.02 \times 10^{10}$ | 35,000 [17] | $2.95 \times 10^{15}$ | $1.26 \times 10^{15}$ | $3.16 \times 10^{15}$ | $2.46 \times 10^{15}$ |
| Pesticide (yuan) | 1800 | 375 | 2000 | $4.37 \times 10^{11}$ * | $7.87 \times 10^{14}$ | $1.64 \times 10^{14}$ | $8.74 \times 10^{14}$ | $6.08 \times 10^{14}$ |
| Subtotal | | | | | $1.10 \times 10^{16}$ | $7.04 \times 10^{15}$ | $1.24 \times 10^{16}$ | $1.02 \times 10^{16}$ |
| Purchased resource (F),F = F$_R$ + F$_N$ | | | | | $1.73 \times 10^{16}$ | $9.85 \times 10^{15}$ | $1.91 \times 10^{16}$ | $1.54 \times 10^{16}$ |
| Total input (U), U = I + F | | | | | $2.01 \times 10^{16}$ | $1.26 \times 10^{16}$ | $2.19 \times 10^{16}$ | $1.82 \times 10^{16}$ |
| **Yield(Y$_1$)** | 13,200 | 8250 | 16,500 | $1.21 \times 10^{12}$ | $1.60 \times 10^{16}$ | $9.98 \times 10^{15}$ | $2.00 \times 10^{16}$ | $1.53 \times 10^{16}$ |
| Ecological benefits (Y$_2$) | | | | | | | | |
| Water conservation (W$_C$) | $3.01 \times 10^{10}$ | $3.01 \times 10^{10}$ | $3.01 \times 10^{10}$ | 23,500 | $7.07 \times 10^{14}$ | $7.07 \times 10^{14}$ | $7.07 \times 10^{14}$ | $7.07 \times 10^{14}$ |
| Soil reinforcement (S$_R$) | | | | | | | | |

Table A1. *Cont.*

| Item | Raw Amounts | | | EUVs (sej unit$^{-1}$) | Solar Emergy (sej) | | | Average (sej) |
|---|---|---|---|---|---|---|---|---|
| | 1 | 2 | 3 | | 1 | 2 | 3 | |
| Total N (g) | 902 | 902 | 902 | $4.64 \times 10^8$ [18] | $4.18 \times 10^{11}$ | $4.18 \times 10^{11}$ | $4.18 \times 10^{11}$ | $4.18 \times 10^{11}$ |
| Total P (g) | 799 | 799 | 799 | $5.07 \times 10^9$ [18] | $4.05 \times 10^{12}$ | $4.05 \times 10^{12}$ | $4.05 \times 10^{12}$ | $4.05 \times 10^{12}$ |
| Total K (g) | 20,300 | 20,300 | 20,300 | $1.31 \times 10^9$ | $2.66 \times 10^{13}$ | $2.66 \times 10^{13}$ | $2.66 \times 10^{13}$ | $2.66 \times 10^{13}$ |
| Organic (J) | $1.95 \times 10^8$ | $1.95 \times 10^8$ | $1.95 \times 10^8$ | 94,100 [66] | $1.83 \times 10^{13}$ | $1.83 \times 10^{13}$ | $1.83 \times 10^{13}$ | $1.83 \times 10^{13}$ |
| subtotal | | | | | $4.94 \times 10^{13}$ | $4.94 \times 10^{13}$ | $4.94 \times 10^{13}$ | $4.94 \times 10^{13}$ |
| Fertility ($F_E$) | | | | | | | | |
| Total N (g) | 600,000 | 493,000 | 606,000 | $4.64 \times 10^8$ [18] | $2.78 \times 10^{14}$ | $2.29 \times 10^{14}$ | $2.81 \times 10^{14}$ | $2.63 \times 10^{14}$ |
| Total P (g) | 920,000 | 385,000 | 971,000 | $5.07 \times 10^9$ [18] | $4.66 \times 10^{15}$ | $1.95 \times 10^{15}$ | $4.92 \times 10^{15}$ | $3.85 \times 10^{15}$ |
| Total K (g) | 169,000 | 25,600 | 119,000 | $1.31 \times 10^9$ | $2.22 \times 10^{14}$ | $3.35 \times 10^{13}$ | $1.55 \times 10^{14}$ | $1.37 \times 10^{14}$ |
| Organic (J) | $1.43 \times 10^{10}$ | $1.43 \times 10^{10}$ | $1.43 \times 10^{10}$ | 94,100 [66] | $1.35 \times 10^{15}$ | $1.35 \times 10^{15}$ | $1.35 \times 10^{15}$ | $1.35 \times 10^{15}$ |
| subtotal | | | | | $6.51 \times 10^{15}$ | $3.56 \times 10^{15}$ | $6.71 \times 10^{15}$ | $5.59 \times 10^{15}$ |
| Carbon fixation ($C_F$) (g) | 17,500,000 | 13,800,000 | 17,000,000 | 6,190,000 [18] | $1.09 \times 10^{14}$ | $8.53 \times 10^{13}$ | $1.05 \times 10^{14}$ | $9.97 \times 10^{13}$ |
| Oxygen production ($O_P$) (g) | 14,300,000 | 16,900,000 | 20,800,000 | 1,220,000 [18] | $1.75 \times 10^{13}$ | $2.06 \times 10^{13}$ | $2.54 \times 10^{13}$ | $2.12 \times 10^{13}$ |
| Total $Y_2 = W_C + S_R + F_E + C_F + O_P$ | | | | | $7.39 \times 10^{15}$ | $4.43 \times 10^{15}$ | $7.59 \times 10^{15}$ | $6.47 \times 10^{15}$ |
| **Indices** | | | | | | | | |
| Em-Power Density (EPD) | | | | | $2.01 \times 10^{12}$ | $1.26 \times 10^{12}$ | $2.19 \times 10^{12}$ | $1.82 \times 10^{12}$ |
| Emergy Self-sufficiency Ratio (ESR) | | | | | 0.14 | 0.22 | 0.13 | 0.15 |
| Emergy Exchange Ratio (EER) | | | | | 0.86 | 0.88 | 0.79 | 0.84 |
| Emergy Yield Ratio (EYR) | | | | | 0.92 | 1.01 | 1.04 | 0.99 |
| Environmental Loading Ratio (ELR) | | | | | 1.25 | 1.32 | 1.34 | 1.30 |
| Emergy Restoration Ratio (ERR) | | | | | 0.43 | 0.45 | 0.40 | 0.42 |
| Emergy Benefit Ratio (EBR) | | | | | 1.35 | 1.46 | 1.44 | 1.41 |
| Emergy Sustainability Index (ESI) | | | | | 0.74 | 0.77 | 0.78 | 0.76 |
| Emergy Index for Sustainable Development (EISD) | | | | | 0.63 | 0.67 | 0.62 | 0.63 |

* EMR (Emergy Money Ratio) was deduced by the linear correlation between the emergy/money ratio and GDP as the Chinese GDP smoothing index of year 2005–2013 is 2.155 and converted to 12.0 E24 sej year$^{-1}$ baseline from 9.26 E24 sej year$^{-1}$.

**Table A2.** Economical raw amounts of the CP in 2017 (/ha/year).

| Iterm | Money (yuan) | | | Average (yuan) |
|---|---|---|---|---|
| | 1 | 2 | 3 | |
| **Input(I)** | | | | |
| **Input for actual (I$_a$)** | | | | |
| Labor of purchased | 0.00 | 0.00 | 0.00 | 0.00 |
| Seeds of corn | 825.00 | 750.00 | 825.00 | 800.00 |
| Compound fertilizer | 3300.00 | 0.00 | 4000.00 | 2433.33 |
| Nitrogenous fertilizer | 0.00 | 1012.50 | 0.00 | 337.50 |
| Pesticide | 1800.00 | 375.00 | 2000.00 | 1391.67 |
| Subtotal I$_a$ | 5925.00 | 2137.50 | 6825.00 | 4962.50 |
| **Input for free (I$_f$)** | | | | |
| Labor of farmers | 3000.00 | 8000.00 | 1800.00 | 4266.67 |
| Manure | 21,000.00 | 9000.00 | 22,500.00 | 17,500.00 |
| Subtotal I$_f$ | 24,000.00 | 17,000.00 | 24,300.00 | 21,766.67 |
| **Total input I = I$_a$ + I$_f$** | 29,925.00 | 19,137.50 | 31,125.00 | 26,729.17 |
| **Output (O)** | | | | |
| Corn | 13,200.00 | 8250.00 | 16,500.00 | 12,650.00 |
| **Total output** | 13,200.00 | 8250.00 | 16,500.00 | 12,650.00 |
| **Indices** | | | | |
| **Input with I$_f$** | 29,925.00 | 19,137.50 | 31,125.00 | 26,729.17 |
| **Input without I$_f$** | 5925.00 | 2137.50 | 6825.00 | 4962.50 |
| **Output** | 13,200.00 | 8250.00 | 16,500.00 | 12,650.00 |
| **Economic Output/Input with If (O/I)** | 0.44 | 0.43 | 0.53 | 0.47 |
| **Economic Output/Input without If (O/I$_a$)** | 2.23 | 3.86 | 2.42 | 2.84 |
| **Economic benefit per unit (EBU, O-I)** | −16,725.00 | −10,887.50 | −14,625.00 | −14,079.17 |
| **Economic pure benefit per unit (EPBU, I$_a$)** | 7275.00 | 6112.50 | 9675.00 | 7687.50 |

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
