# Peer review of "Emergy and Economic Evaluation of Seven Typical Agroforestry Planting Patterns in the Karst Region of Southwest China"

_forests, doi:10.3390/f10020138_

Round 1

Reviewer 1 Report

In the data collection section, details of variables covered (basic economic data and agriculture performance data) in the questionnaire survey are required.

In the data analysis section, details are required on what data feed into the different emergy and economic parameters. For example, how were the values in Fig 1 were obtained and how do they fit into emergy and economic models.

Author Response

We would like to thank you first for all the positive comments of our manuscript Manuscript ID: forests-438903 entitled “Emergy and economic evaluation of seven typical agroforestry planting 
patterns in the karst region of Southwest China”. We really appreciate your help and patience.

1. In the data collection section, details of variables covered (basic economic data and agriculture performance data) in the questionnaire survey are required.

Yes, after our discussion, two appendix files are added at the end of the article. Appendix A is emergy analysis and appendix B is economic analysis. The first columns of these tables are categories of basic economic data and agricultural performance data, and the second to fourth columns are raw data.

2. In the data analysis section, details are required on what data feed into the different emergy and economic parameters. For example, how were the values in Fig 1 were obtained and how do they fit into emergy and economic models.

Yes, after we added the appendix, the calculation process of emergy analysis is clear. The data in Fig 1.a is taken from appendix A. The unit emergy data used also indicates the source and is added to the references section.

Reviewer 2 Report

Summary and overall review

The paper analyse the economic and ecologic benefits of the seven most typical agroforestry systems in the Yunnan Province, China. The study utilise emergy analysis to quantify the increased benefits of agroforestry production systems, and highlight the need for including a focus on ecological as well as economic benefits of agroforestry systems in policy-making.

The study is original and contributes important information on sustainability of agroforestry. Thus, it is of great interest to a broad range of academics, policy-makers and farmers. The method is succinct and the results are presented in a professional and approachable manner. The study is supported by relevant references of high quality, and conclusion and recommendations are clearly expressed and well supported by robust results.  

The manuscript follows a natural flow with a clear distinction between sections. The objectives of the study is clearly outlined in the introduction, and the following sections is written with that clear aim in mind, which leads to well supported conclusion that addresses all objectives considerately. It is approachable, concise and exhibits and high level of the English language.

Congratulations on a well-carried out study of great importance and interest!

General corrections

I suggest to clarify that the following sections of 2.2. (Line 108) are descriptions of the systems. E.g. “2.2. Description of the seven agroforestry planting patterns”

Remove full stop at the end of each headline in Method (Line 109, 114, 123, 130, 138, 142, 149, 175, 207, and 218).

In Figure 1, please specify in figure text which diagram depicts which production system (Line 251).

In Table 4 (Line 263), please fit table to page margin. If necessary, text size can be reduced to the same size as the table description text. However, in that case, the text size should be adjusted to the same size in all tables.

In Table 5 (Line 293), please explain abbreviation in figure text (I, O, If and Ia). Sub-script ‘f’ in ‘If’ in the 2nd and 6th row. Remove multiply sign from units in the first column. Add thousands separator in numbers. There is an extra ‘u’ in ‘Unit net economic benefit (EPBU)’.

Please add full stop at the end of every table and figure text.

Please elaborate shortly on where Castellini’s study was conducted (Line 396 in Discussion).  

Specific comments

1.       Line 36 in Abstract: Remove comma after ‘both’.

2.       Line 61 in Introduction: Remove comma after ‘e.g.’

3.       Line 86 in Introduction: Switch ‘also’ and have’. “…People of Yunnan Province have also implemented…”

4.       Line 159 in section 2.3: Add plural ‘s’ to ‘survey’.

5.       Line 108 in section 2.4.1: Remove comma after ‘components’. Changes “materials and money and energy flows” to “materials, money and energy flows”.

6.       Line 181: the ‘m’ in ‘emergy’ appears to be of a different text font.

7.       Line 182: Add ‘.’ after ‘etc’ in parenthesis.

8.       Line 188: Replace comma with ‘and’ in “…water, seed”.

9.       Line 189: Remove multiply dot in the unit.

10.   Line 234 in Results: Change ‘goat’ to ‘sheep’.

11.   Line 250 in Figure 1: Correct system name of diagram ‘g’ from “Pomegranate system” to “Pomegranate-grass-sheep system”.

12.   Line 264: Please add one line of space between table and text.

13.   Line 280: Add ‘was’ between ‘which’ and ‘also’ - > “…which was also higher than…”

14.   Line 294: Please add one line of space between table and text.

15.   Line 294: Remove multiply sign in unit in parenthesis.

16.   Line 326 Table 6: Please add thousands separator.

17.   Line 342 in Discussion: Consider to replace ‘or’ with ‘and’ for clarification.

18.   Line 375: Change first letter in ‘Maize, Wheat, Rice and Soybean’ to non-capitals.

19.   Line 377: Remove comma after ‘other’.

20.   Line 382: Replace comma between ‘high-input’ and ‘high-out’ with ‘and’.

21.   Line 410: Change “it is still need to” to “there is still a need to”

22.   Line 413: Replace ‘counted with’ with ‘included’.

23.   Line 422 in Conclusion: Please spell out ‘6’ as ‘six’.

24.   Line 464 in References: Is the journal abbreviated correctly?   

25.   Line 518. Please abbreviate the name of the journal.

26.   Line 525, 557, 559, 577, 585, 590 and 597: Please remove ‘&’ in between last and 2nd last author.

Author Response

We would like to thank you first for all the positive comments of our manuscript Manuscript ID: forests-438903) entitled “Emergy and economic evaluation of seven typical agroforestry planting 
patterns in the karst region of Southwest China”. We really appreciate your help and patience.

1.       I suggest to clarify that the following sections of 2.2. (Line 108) are descriptions of the systems. E.g. “2.2. Description of the seven agroforestry planting patterns”

Yes, the revised section can be seen in Line 109.

2.       Remove full stop at the end of each headline in Method (Line 109, 114, 123, 130, 138, 142, 149, 175, 207, and 218).

Yes, the revised section can be seen in Line 111, 116, 125, 132, 140, 144, 151, 177, 210, 221.

3.       In Figure 1, please specify in figure text which diagram depicts which production system (Line 251).

Yes, the information added can be seen in Line 255-257.

4.       In Table 4 (Line 263), please fit table to page margin. If necessary, text size can be reduced to the same size as the table description text. However, in that case, the text size should be adjusted to the same size in all tables.

Yes, we have corrected the problem in line 269.

5.       In Table 5 (Line 293), please explain abbreviation in figure text (I, O, If and Ia). Sub-script ‘f’ in ‘If’ in the 2nd and 6th row. Remove multiply sign from units in the first column. Add thousands separator in numbers. There is an extra ‘u’ in ‘Unit net economic benefit (EPBU)’.

Yes, the information added can be seen in Line 300-303. The thousand separator has been added to Table 5. Spelling errors have also been corrected.

6.       Please add full stop at the end of every table and figure text.

Yes, we have corrected the problem.

7.       Please elaborate shortly on where Castellini’s study was conducted (Line 396 in Discussion). 

 Yes, the information added can be seen in Line 406.

Specific comments

1.                        Line 36 in Abstract: Remove comma after ‘both’.

Yes, we have corrected the problem in line 37.

2.                        Line 61 in Introduction: Remove comma after ‘e.g.’

Yes, we have corrected the problem in line 62.

3.                        Line 86 in Introduction: Switch ‘also’ and have’. “…People of Yunnan Province have also implemented…”

Yes, the revised section can be seen in Line 86.

4.                        Line 159 in section 2.3: Add plural ‘s’ to ‘survey’.

Yes, we have corrected the problem in line 161.

5.                        Line 108 in section 2.4.1: Remove comma after ‘components’. Changes “materials and money and energy flows” to “materials, money and energy flows”.

Yes, the revised section can be seen in Line 181.

6.                        Line 181: the ‘m’ in ‘emergy’ appears to be of a different text font.

Yes, we have corrected the problem in line 182.

7.                        Line 182: Add ‘.’ after ‘etc’ in parenthesis.

Yes, we have corrected the problem in line 184.

8.                        Line 188: Replace comma with ‘and’ in “…water, seed”.

Yes, we have corrected the problem in line 190.

9.                        Line 189: Remove multiply dot in the unit.

Yes, the revised section can be seen in Line 192.

10.                    Line 234 in Results: Change ‘goat’ to ‘sheep’.

Yes, we have corrected the problem in line 237.

11.                    Line 250 in Figure 1: Correct system name of diagram ‘g’ from “Pomegranate system” to “Pomegranate-grass-sheep system”.

Yes, We have redrawn Figure 1.

12.                    Line 264: Please add one line of space between table and text.

Yes, we have corrected the problem.

13.                    Line 280: Add ‘was’ between ‘which’ and ‘also’ - > “…which was also higher than…”

Yes, the revised section can be seen in Line 192.

                14.   Line 294: Please add one line of space between table and text.

Yes, the revised section can be seen in Line 286.

14.                    Line 294: Remove multiply sign in unit in parenthesis.

Yes, we have corrected the problem.

15.                    Line 326 Table 6: Please add thousands separator.

Yes, the thousand separator has been added to Table 6.

16.                    Line 342 in Discussion: Consider to replace ‘or’ with ‘and’ for clarification.

Yes, we have corrected the problem.

17.                    Line 375: Change first letter in ‘Maize, Wheat, Rice and Soybean’ to non-capitals.

Yes, we have corrected the problem in 385.

18.                    Line 377: Remove comma after ‘other’.

Yes, we have corrected the problem in line 388.

19.                    Line 382: Replace comma between ‘high-input’ and ‘high-out’ with ‘and’.

Yes, we have corrected the problem in line 393.

20.                    Line 410: Change “it is still need to” to “there is still a need to”

Yes, the revised section can be seen in Line 421.

21.                    Line 413: Replace ‘counted with’ with ‘included’.

Yes, the revised section can be seen in Line 424.

22.                    Line 422 in Conclusion: Please spell out ‘6’ as ‘six’.

Yes, the revised section can be seen in Line 433.

23.                    Line 464 in References: Is the journal abbreviated correctly?  

Yes, We have corrected this abbreviation in line 495.

24.                    Line 518. Please abbreviate the name of the journal.

Yes, we've abbreviated it in line 544.

25.                    Line 525, 557, 559, 577, 585, 590 and 597: Please remove ‘&’ in between last and 2nd last author.

Yes, we have corrected the problem.
